# High Rate of Cytokine Release Syndrome-Related Coagulopathy with Low Incidence of Bleeding and Thrombosis in Patients Treated with B-Cell Maturation Antigen (BCMA)-Targeted Chimeric Antigen Receptor T-Cells (CAR-T)

**DOI:** 10.3390/cancers17213551

**Published:** 2025-11-02

**Authors:** Ariela Arad, Maya Katz, Eyal Lebel, Yosef Kalish, Miri Assayag, Batia Avni, Shlomo Elias, Sigal Grisariu, Ela Shai, Shlomit Kfir-Erenfeld, Nathalie Asherie, Moshe E. Gatt, Polina Stepensky, Eran Zimran

**Affiliations:** 1Department of Hematology, Hadassah Hebrew University Medical Center, Jerusalem 91200, Israel; 2Faculty of Medicine, Hebrew University of Jerusalem, Jerusalem 91904, Israel; 3Department of Bone Marrow Transplantation and Cancer Immunotherapy, Hadassah Hebrew University Medical Center, Jerusalem 91200, Israel

**Keywords:** cytokine release syndrome, B-cell maturation antigen (BCMA), chimeric antigen receptor T-cell (CAR-T), coagulopathy, thrombosis, bleeding

## Abstract

**Simple Summary:**

Chimeric antigen receptor T-cell (CAR-T) therapy targeting B-cell maturation antigen is a promising treatment for patients with advanced multiple myeloma and AL amyloidosis. Cytokine release syndrome (CRS) is the most common adverse event associated with CAR-T treatment; however, some of its accompanying conditions are poorly characterized. One such condition is a clotting abnormality termed coagulopathy that may result in serious bleeding or thrombosis. We sought to characterize this coagulopathy in a cohort of 108 patients who received CAR-Ts in a clinical trial. CRS occurred in 93% of the patients, with some extent of coagulopathy in 73% and severe coagulopathy in 19%. Despite the high rate of coagulopathy, thrombosis (4.6%) and bleeding (2.7%) events were rare and unrelated to the degree of CRS or coagulopathy. Furthermore, no serious bleeding or fatal thrombosis occurred. These outcomes appear comparable or superior to those in previous studies, suggesting the benefit of the prophylactic measures used.

**Abstract:**

**Background**: B-cell maturation antigen (BCMA)-targeted chimeric antigen receptor T-cell (CAR-T) therapy has demonstrated substantial efficacy in relapsed and/or refractory multiple myeloma. While toxicities such as cytokine release syndrome (CRS) and immune effector cell-associated neurotoxicity syndrome (ICANS) have been well characterized, the incidence and clinical consequences of the coagulopathy associated with CRS remain underexplored. **Methods**: We conducted a prospective analysis of 108 adult patients with multiple myeloma or light chain amyloidosis treated with the academic anti-BCMA CAR-T HBI0101 in a single-center trial (NCT04720313). Coagulopathy was evaluated via serial fibrinogen measurements, with hypofibrinogenemia defined as <200 mg/dL and severe coagulopathy as <100 mg/dL. Laboratory markers, tocilizumab and blood product use, and thrombotic and bleeding complications were recorded. Patients received a short (3-day) or extended course of enoxaparin thromboprophylaxis as well as fresh frozen plasma in cases of severe coagulopathy. **Results**: CRS grades 1–3 occurred in 100 patients (93%). Hypofibrinogenemia was observed in 79 patients (73%), including 20 (19%) with severe coagulopathy. Fibrinogen levels were significantly associated with CRS severity (*p* < 0.001), number of tocilizumab doses (*p* < 0.001), peak levels of the inflammation markers LDH (*p* = 0.001) and ferritin (*p* = 0.006), and neutropenia (*p* = 0.33). Five thrombotic events (4.6%) and three minor bleeding events (2.7%) occurred within 3 months post-CAR-T infusion and were not associated with degree of coagulopathy or CRS. No cases of major bleeding or fatal thrombosis occurred. **Conclusions**: CRS-related coagulopathy is common following BCMA-targeted CAR-T treatment and correlates closely with CRS severity. Despite the high rate of laboratory coagulopathy, thrombosis and bleeding events were infrequent, suggesting the benefit of the prophylactic strategies used.

## 1. Introduction

B-cell maturation antigen (BCMA)-targeted chimeric antigen receptor T-cell (CAR-T) therapy has emerged as a major breakthrough in the treatment of relapsed/refractory (RR) multiple myeloma (MM), offering a high response rate even in high-risk patients [1,2,3,4,5,6]. A novel, locally produced, BCMA-targeted CAR-T for the treatment of RR-MM (HBI0101) was developed in our institute and evaluated in a phase-1a/b study, between March 2021 and January 2025 (NCT04720313) [7]. We previously reported manageable toxicities and high efficacy with HBI0101, in a frailer and higher-risk MM patient population as compared with other anti-BCMA CAR-T studies, including an overall response rate of 92% and a median progression-free survival of 11.6 months [8]. Remarkably, this study was the first to evaluate the use of CAR-T for the treatment of patients with light chain (AL) amyloidosis [9].

The most common toxicity associated with CAR-T treatment is a systemic inflammatory response termed cytokine release syndrome (CRS), which is characterized by high fever, tachycardia, hypotension, hypoxia, and organ dysfunction [10,11,12,13,14]. A diffuse intravascular coagulation (DIC)-like coagulopathy manifesting as prolonged PT and aPTT, hypofibrinogenemia, and thrombocytopenia has been previously reported in patients treated with CD19-targeted CAR-T for RR B-cell acute lymphoblastic leukemia (B-ALL) and large B-cell lymphoma (LBCL) and was shown to correlate with CRS. This so called CRS-related coagulopathy has occasionally been associated with a high rate of bleeding and thrombosis of up to 19% and 11%, respectively [15,16]. However, little is known about the incidence of CRS-related coagulopathy following CAR-T in patients with RR-MM and AL amyloidosis, and its associated clinical consequences.

The association between inflammation and thrombosis has been increasingly recognized in recent years. The term “thromboinflammation” was first used by Blair et al. in 2009 to describe the activation of Toll-like receptors on platelets, leading to both inflammatory response and thrombosis [17]. CRS is characterized by elevated levels of inflammatory cytokines, including interleukin-6 (IL-6), interleukin-1 (IL-1), interferon-gamma (IFN-γ), and tumor necrosis factor-alpha (TNF-α) [18]. Microvascular thrombosis has been reported in fatal cases of severe CAR-T-related toxicity [19]. Furthermore, thrombosis is a common complication of MM, which predisposes patients to a hypercoagulable state due to multiple factors including secretion of proinflammatory cytokines, increased platelet activation, and the use of pro-thrombotic treatments [20,21]. Hence, exposure of patients with advanced and active MM to the potential thromboinflammation associated with CRS raises concern for a high risk of thrombotic complications.

In this study, we aimed to characterize the frequency, laboratory findings, and clinical correlations associated with CRS-related coagulopathy in adult patients with RR-MM and AL amyloidosis receiving BCMA-targeted CAR-T.

## 2. Subjects and Methods

### 2.1. Study Population

This study included all patients who were enrolled between March 2021 and February 2024 in an ongoing phase I-a/b clinical trial evaluating HBI0101, an academic, locally produced, BCMA-targeted CAR-T for the treatment of RR-MM and AL amyloidosis (NCT04720313). The trial was conducted at the Department of Hematology and the Department of Bone Marrow Transplantation and Cancer Immunotherapy, Hadassah Medical Center, Jerusalem, Israel. A detailed description of the study protocol has been reported previously [7,8,9]. Briefly, the key eligibility criteria required three or more previous lines of therapy, including a proteasome inhibitor, an immunomodulatory agent, and an anti-CD38 antibody. Criteria with regards to organ function and hematological reserve were relatively permissive, with a minimum platelet count of 30 × 10^9^/L, creatinine clearance of 20 mL/min, left ventricular ejection fraction of 40%, and performance status of 0–2 on the ECOG scale. Lymphodepletion chemotherapy consisted of fludarabine 25 mg/m^2^ once daily and cyclophosphamide 250 mg/m^2^ once daily on days −5 to −3 before CAR-T infusion. Bendamustine 90 mg/m^2^ once daily on days −4 and −3 was administered to patients with a creatinine clearance of < 30 mL/min. Of note, three patients with poor performance status (ECOG ≥ 3) who were excluded from the clinical trial but received HBI0101 via a compassionate program were also included in this analysis. The trial and compassionate program were both approved by the institutional ethics committee according to the Declaration of Helsinki, and all patients provided written informed consent, which included anonymized data collection.

### 2.2. Evaluation of Laboratory and Clinical Parameters

Complete blood count (CBC) and coagulation parameters, including prothrombin time (PT), activated partial thromboplastin time (aPTT), and fibrinogen levels, were measured before lymphodepletion, at least three times per week during the first two weeks following CAR-T administration and subsequently as determined by the treating physician. Coagulopathy was defined as a fibrinogen nadir of ≤200 mg/dL. A threshold fibrinogen of 100 mg/dL was used to differentiate between severe and mild or moderate coagulopathy. These cutoff values were defined based on the recently published International Society on Thrombosis and Haemostasis (ISTH) definition and scoring of DIC [22], as well as observations linking fibrinogen levels ≤ 200 mg/dL to a higher risk of bleeding [23,24]. Despite the lack of D-dimer measurements in the study protocol, overt DIC was approximated using the above criteria [22]. CRS and immune effector cell-associated neurological toxicity (ICANS) were graded according to the 2019 ASTCT consensus grading [25]. Immune effector cell-associated hemophagocytic lymphohistiocytosis-like syndrome (IEC-HS) was defined and graded according to the ASTCT schema [26]. Thrombotic events were defined according to ICD-10 and categorized as either venous thromboembolism (VTE) or arterial events. Bleeding events were defined according to the ISTH scoring system [27].

### 2.3. Thromboprophylaxis and Blood Product Support

All patients were managed with a peripherally inserted central catheter (PICC) during the hospitalization for CAR-T. Due to the estimated high risk of VTE in this patient population [20,21], all patients received prophylactic enoxaparin at a dose of 0.5 mg/kg for a minimum of three days following PICC insertion. In patients considered to be at an especially high risk of thrombosis (i.e., history of VTE, thrombophilia, obesity, high MM disease burden, or masses in proximity to blood vessels) prophylactic enoxaparin was continued until platelet counts fell below 50 × 10^9^/mL or throughout the hospital stay. Fresh frozen plasma (FFP) at a dose of 5–10 mL/kg was administered prophylactically once daily or once every other day (q2d) in cases of severe coagulopathy (fibrinogen < 100 mg/dL). Cryoprecipitate was restricted for cases of extremely low fibrinogen levels of <40 mg/dL or any coagulopathy associated with clinically significant bleeding. Platelet transfusions were administered prophylactically in cases of platelet count decrease to <10 × 10^9^/L or grade-3 thrombocytopenia (<50 × 10^9^/L) associated with clinically significant bleeding. In patients who received extended enoxaparin prophylaxis and developed severe coagulopathy without grade-3 thrombocytopenia, enoxaparin was administered concurrently with FFP, or held, according to decision of the treating physicians.

### 2.4. Statistical Analysis

Descriptive statistics included the mean ± standard deviations for continuous variables. Chi-square or Fisher’s exact tests were used for comparison between categorical variables as appropriate. The Wilcoxon test or the Mann–Whitney test was used for comparison between continuous variables, as appropriate. All reported *p* values were two-sided and were judged to be significant at less than 0.05. The results were analyzed, and figures were produced using IBM SPSS Statistics version 30 software.

## 3. Results

### 3.1. Patient Characteristics

Baseline patient characteristics are summarized in Table 1. A total of 108 patients were included in the analysis, with a mean age of 64 years (range 35–84) and a slight female predominance (52%). Most of the patients had MM (*n* = 95), while a small subset had AL amyloidosis (n = 13). The median number of prior lines of therapy was four, consistent with a heavily pretreated patient population. Common comorbidities included hypertension (29%), dyslipidemia (25%), type 2 diabetes mellitus (15%), and smoking (23%). Furthermore, 43% of patients had at least one cardiovascular risk factor, 16% had a history of ischemic heart disease or stroke, and 15% had a history of VTE.

### 3.2. CAR-T-Associated Toxicities

Overall, 100 patients (93%) experienced CRS of any grade, including 36 (33%) with grade-1, 49 (45%) with grade-2, and 15 (14%) with grade-3 CRS. There were no cases of grade-4 or fatal CRS. Furthermore, 82 of the above 100 patients received treatment with at least one dose of the IL-6 antagonist tocilizumab. Six patients experienced ICANS, all low grade, while three patients experienced IEC-HS of grades 1 or 2.

The associations between patient variables and degree of coagulopathy are summarized in Table 2. Using fibrinogen levels as a primary marker of coagulopathy [19], 79 patients (73%) developed coagulopathy including 59 (54%) with fibrinogen 100–200 mg/dL and 20 (19%) with fibrinogen < 100 mg/dL. Patients with severe coagulopathy tended to be older (mean age 68.5 years as compared with 63.3 and 61.8 in patients with fibrinogen levels of 100–200 and > 200 mg/dL, respectively) and had a higher prevalence of cardiovascular risk factors (50% vs. 44% and 37%) and history of VTE (25% vs. 15% and 7%). Coagulopathy lagged after the onset of CRS. An initial decrease in fibrinogen levels was noted at a median of 3 days post-CRS onset (range 2–5), while nadir fibrinogen levels were reached at a median of 7 days post-CRS onset (range 6–11).

Consistent with DIC-like coagulopathy, lower fibrinogen levels correlated significantly with lower platelet counts (*p* = 0.044) and higher levels of PT/INR (*p* < 0.001). Of the 20 patients who developed severe coagulopathy, 9 fulfilled ISTH criteria for overt DIC using only the degree of PT/INR elevation and thrombocytopenia in addition to fibrinogen <100 mg/dL. Since D-dimer measurements were not available, this may be an underestimation.

### 3.3. The Association Between CRS and Coagulopathy

Coagulopathy was strongly associated with CRS severity. All patients who developed coagulopathy had CRS of at least grade-1, and over half of the patients with fibrinogen levels <100 mg/dL had grade-2 or -3 CRS (*p* < 0.001) (Figure 1A, Table 2). A similar association was observed between CRS grade and abnormally elevated PT/INR (Figure 1B). Patients who received more tocilizumab doses, which served as a surrogate for CRS severity, were more likely to have fibrinogen <100 mg/dL (*p* < 0.001) (Figure 1C). Furthermore, severe coagulopathy was significantly associated with higher peak levels of CRS-related markers of inflammation, LDH (*p* = 0.001) and ferritin (*p* = 0.006), but not with CRP. Notably, severe coagulopathy was also associated with lower neutrophil counts post-CAR-T treatment (*p* = 0.33, Table 2). All three patients who experienced IEC-HS concurrently exhibited severe coagulopathy and had grade-2 or -3 CRS. In contrast, we observed no apparent association between ICANS and coagulopathy.

### 3.4. Thrombosis and Bleeding Outcomes

Table 3 provides a case-by-case detail of thrombosis and bleeding events. Five patients (4.6%) experienced thrombotic events within 3 months following CAR-T infusion, which included two cases of non-fatal acute myocardial infarction and three cases of VTE. All five patients had history of cardiovascular comorbidities and/or VTE. Two of five events occurred post-discharge, following resolution of the laboratory abnormalities associated with coagulopathy. These events could be attributed to active MM, as the above two patients achieved a response to CAR-T. Remarkably, there were no major or clinically relevant non-major bleeding events. Three minor bleeding events (2.7%) included mild hematuria, epistaxis, and cutaneous bleeding occurred, each in one patient. Median onset of thrombotic or minor bleeding events was 13 days post-infusion (range: 3–30 days). No associations between thrombosis or bleeding events and fibrinogen levels, PT/INR, aPTT, or platelet counts were observed. Similarly, no association was observed between the severity of CRS and thrombosis or bleeding events; however, the statistical analysis lacks power due to the small number of events.

Interestingly, while the frequency of extramedullary disease was 24/95 (25%) among the patients with MM; six of an overall eight cases (75%) of thrombosis or bleeding events occurred in patients with extramedullary MM. None of the events directly involved the extramedullary mass. Also of note, none of the thrombosis or minor bleeding events occurred among the 13 patients with AL amyloidosis included in the study.

Approximately one-third of the patients received prophylactic enoxaparin throughout the hospitalization or until their platelet count decreased to <50 × 10^9^/mL. Eighteen patients required one or more administrations of prophylactic FFP, most of them (*n* = 15) due to severe coagulopathy as per protocol. A higher proportion of patients with severe coagulopathy required administration of packed red blood cells (*p* = 0.029) and platelet transfusions (*p* = 0.06). Notably, cryoprecipitate was not used in patient management.

## 4. Discussion

In this single-center cohort of 108 patients with RR-MM and AL amyloidosis treated with a locally produced academic BCMA-targeted CAR-T, we observed a high rate (73%) of CRS-related coagulopathy. This DIC-like coagulopathy, defined by abnormally low fibrinogen levels, was closely correlated with the severity of CRS, number of tocilizumab doses, markers of inflammation, and, interestingly, the degree of neutropenia. Within a follow-up of three months post-CAR-T infusion, five patients (4.6%) experienced arterial or venous thrombotic events, and there were no major or clinically relevant non-major bleeding events. We did not observe an apparent association between thrombotic or minor bleeding events and the degree of coagulopathy or CRS; however, the low number of events limits statistical analysis.

Several studies have contributed to the characterization of CRS-related coagulopathy in CD19-targeted CAR-T therapy; however, its pathogenesis has not been clearly elucidated. Proposed mechanisms are liver injury resulting in decreased production of coagulation factors and endothelial damage with release of von Willebrand Factor [28]. In a recent study in patients with RR-LBCL, a transient hypofibrinolytic state during CRS was observed leading to mild hypercoagulability as evidenced by increased soluble fibrin and thrombin–antithrombin complexes; however, no thrombotic or bleeding events were observed [29]. Other studies reported inconsistent rates of thrombosis and bleeding events post-CAR-T. In a study of 127 patients who received CAR-T for B-ALL and LBCL, 9.4% developed thrombosis and 6.3% experienced bleeding within three months post-CAR-T. Bleeding events were linked to older age, lower baseline platelet counts, reduced platelet and fibrinogen nadirs, and elevated LDH [30]. In a study of 148 patients receiving CAR-T for LBCL, 11% developed a new venous thromboembolism (VTE) within 100 days of infusion, with higher risk observed in those with bulky disease, bridging therapy, poor performance status, severe CRS, and neurotoxicity. Hypofibrinogenemia was observed in some patients but was not linked to major bleeding events [16]. Finally, in a study of 140 patients treated with CAR-T for RR B-ALL, LBCL, and MM, 10 patients (7.1%) experienced thrombotic events, including 9 venous events [31]. Interestingly, the presence of ICANS of any grade was significantly associated with thrombotic events. No significant associations were found with CRS, baseline patient characteristics (e.g., age, BMI, prior history of thrombosis), PT, or aPTT.

In contrast to the evidence in B-ALL and LBCL, only two previous studies addressed coagulopathy in patients with RR-MM treated with BCMA-targeted CAR-T. In a single-center study, coagulation disorders were analyzed in a group of 100 patients with various hematological malignancies receiving CAR-T, including 21 patients with RR-MM. Coagulation abnormalities occurred in 51 patients (51%) in the entire cohort and were associated with more prior lines of therapy, lower baseline platelet counts, and CRS. Clinically significant bleeding occurred in 10 of 51 patients (19.6%). Among 21 patients with MM, only 1 had decreased fibrinogen levels, and 7 had elongated aPTT; however, clinical outcomes were not detailed specifically in these patients [15]. In a subsequent study of 37 patients with RR-MM undergoing BCMA-targeted CAR-T, CRS occurred in all patients, of which 91% developed at least one abnormal coagulation parameter. Coagulation abnormalities closely correlated with severity of CRS. There were no cases of severe bleeding or bleeding involving a critical organ. Minor bleeding and thrombosis outcomes were not detailed [32]. Our findings are consistent with the latter study and confirm its findings in a larger cohort that includes, for the first time, patients with AL amyloidosis.

To our knowledge, this prospective study represents the largest single-center cohort of patients with RR-MM and AL amyloidosis receiving BCMA-targeted CAR-T reported to date and is the first to provide detailed documentation of thrombosis and bleeding complications in this patient population. Despite the high rate of laboratory-defined coagulopathy and a high-thrombotic-risk patient population, incidence of thrombosis and bleeding events was low and appeared to be more strongly associated with preexisting risk factors than with degree of CRS or coagulopathy. We speculate that these low rates of thrombosis and bleeding complications, as compared with previous studies, may be attributed at least in part to the prophylactic strategy used.

Limitations of this study include its single-center and observational design. Furthermore, measurements of D-dimer as well as various markers of coagulation activation (e.g., thrombin generation assays) were not available, precluding an accurate distinction between coagulopathy and ISTH-defined DIC and limiting our ability to address pathophysiological mechanisms. An ongoing larger-scale study will include the above markers along with longer-term follow-up.

## 5. Conclusions

In conclusion, we confirm a high rate of laboratory-defined coagulopathy in a population of patients with RR-MM as well as AL amyloidosis receiving BCMA-targeted CAR-T, closely correlated to CRS severity and markers of inflammation. Despite the high incidence of coagulopathy, the rate of thrombotic and bleeding complications was remarkably low. Use of a short or extended course of enoxaparin thromboprophylaxis in all patients, as well as FFP prophylaxis in cases of severe coagulopathy, appeared beneficial. Further prospective studies are needed to clarify whether routine thromboprophylaxis should be universally recommended for patients with MM and AL amyloidosis undergoing CAR-T therapies.

## Figures and Tables

**Figure 1 cancers-17-03551-f001:**
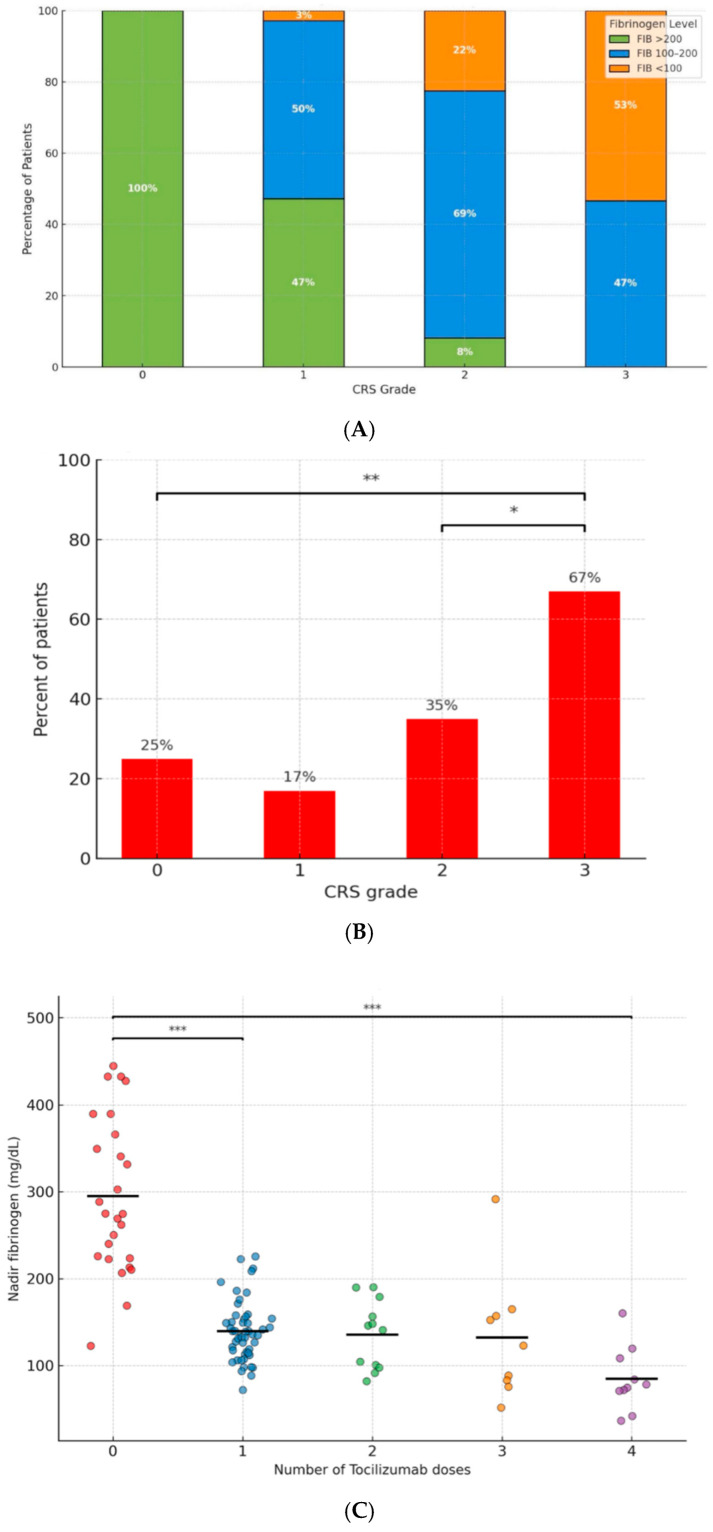
The association between coagulopathy and CRS. (**A**) Stacked bar illustrating the distribution of nadir fibrinogen levels stratified by CRS grade. (**B**) Proportion of patients with abnormal PT/INR values (INR > 1.25) stratified by CRS grade. (**C**) Dot plot showing the association between the number of tocilizumab doses and nadir fibrinogen levels. The horizontal bars represent the median fibrinogen levels within each group (* indicates *p* < 0.05; ** indicates *p* < 0.01; *** indicates *p* < 0.001).

**Table 1 cancers-17-03551-t001:** Patient characteristics.

Variables	Total *n* = 108
Age in years-mean (range)	64 (35–84)
Females, *n* (%)	56 (52%)
Diagnosis: MM/AL	95/13
**Comorbidities, *n* (%)**	
DM2	16 (15%)
Hypertension	31 (29%)
Dyslipidemia	27 (25%)
Obesity	2 (2%)
Smoking	25 (23%)
≥1 cardiovascular risk factors	46 (43%)
IHD (*n* = 16) or stroke (*n* = 1)	17 (16%)
History of VTE	16 (15%)
CKD	2 (2%)
Pulmonary disease	6 (6%)
**ECOG**-**PS, *n* (%)**	
0	11 (10%)
1	46 (43%)
2	48 (44%)
3	2 (2%)
4	1 (1%)
**Disease-related features**	
Previous lines of Tx, median (range)	4 (3–14)
Extramedullary multiple myeloma	24/95 (25%)
High-risk cytogenetics multiple myeloma	42/95 (44%)
Triple-refractory multiple myeloma *	81/95 (85%)
**Lymphodepletion regimen, *n* (%)**	
Fludarabine/cyclophosphamide	102 (94%)
Bendamustine	6 (6%)

Abbreviations: MM: multiple myeloma, AL amyloidosis, DM2: diabetes mellitus type 2; IHD: ischemic heart disease; VTE: venous thromboembolism, CKD: chronic kidney disease. ECOG-PS- performance status by Eastern Cooperative Oncology Group scale; Tx—therapy. * Triple refractory: myeloma refractory to all three classes of novel agents: proteosome inhibitors, immune-modulators, and monoclonal antibodies.

**Table 2 cancers-17-03551-t002:** Association of patient variables with coagulopathy as defined by fibrinogen levels.

	FIB > 200 (mg/dL)	100 < FIB < 200 (mg/dL)	FIB < 100 (mg/dL)	*p* Value
Total patients, *n* = 108 (%)	29 (27%)	59 (54%)	20 (19%)	
Female gender (*n* = 56)	19 (34%)	29 (52%)	8 (14%)	0.18
Age (avg)	61.8	63.3	68.5	**0.047**
**Comorbidities**				
≥1 cardiovascular RF (*n* = 46)	10	26	10	0.52
IHD (n = 16) or stroke (*n* = 1)	7	8	2	0.68
History of VTE (*n* = 16)	2	9	5	0.38
**CRS grade**				
0 (*n* = 8)	8	0	0	**<0.001**
1 (*n* = 36)	17	18	1	**<0.001**
2 (*n* = 49)	4	34	11	**<0.001**
3 (*n* = 15)	0	7	8	**<0.001**
**Tocilizumab doses**				
0 (*n* = 26)	24	2	0	**<0.001**
1 (*n* = 51)	4	41	6	**<0.001**
2 (*n* = 12)	0	9	3	**<0.001**
3 (n = 9)	1	4	4	**<0.001**
4 (*n* = 10)	0	3	7	**<0.001**
**Laboratory values**				
Peak CRP (mg/dL)	8.9	6.5	5.5	0.72
Peak ferritin (ng/mL)	1262	3115	7893	**0.001**
Peak LDH (U/L)	396	423	1082	**0.006**
Nadir neutrophil count (10^9^/L)	0.4	0.4	0.2	**0.033**
Nadir platelet count (10^9^/L)	69	70	41	**0.044**
Nadir hemoglobin (g/dL)	8	8.2	7.9	
Peak PTT (sec)	35.7	34.3	36.2	0.287
Peak PT/INR (sec)	1.1	1.2	1.6	**<0.001**
**Blood product support** (No. of patients receiving ≥ 1 units)				
PLTs	11 (38%)	11 (19%)	8 (40%)	0.06
PCs	16 (55%)	27 (48%)	16 (80%)	**0.029**
FFP	1 (3%)	6 (10%)	15 (75%)	NA
**Thrombosis events**	1	3	1	NA
**Minor bleeding events**	1	2	0	NA

Abbreviations: FIB: fibrinogen; RF: risk factor; IHD: ischemic heart disease; VTE: venous thromboembolism; CRS: cytokine release syndrome; CRP: C-reactive protein; LDH: lactate dehydrogenase; PTT: partial thromboplastin time, PT/INR: prothrombin time/international normalized ratio; PLTs: platelets; FFP: fresh frozen plasma; PCs: packed (red blood) cells; NA: not applicable.

**Table 3 cancers-17-03551-t003:** Cases of arterial and venous thrombotic events and minor bleeding events post-CAR-T treatment.

Case	Age and Gender	Cardiovascular Comorbidities	Myeloma Characteristics	Previous Lines of Tx	EMD	% Plasma Cells in BM	Thrombosis or Bleeding Event	Days post-CAR-T Infusion	CRS Grade	Tocilizumab Doses	Fibrinogen at Event (mg/dL)	Fibrinogen 2 Weeks post-CAR-T Infusion	Fibrinogen 4 Weeks post-CAR-T Infusion	Delta Fibrinogen *
1	58 (F)	HTN, DM2, dyslipidemia	FLC lambda, +1q, t(4:14), del17p	3	Yes	50%	NSTEMI	12	2	3	174	167	152	191
2	61 (F)	CHF	IgG-K, +1q, del13(q)	4	No	40%	DVT	17	1	0	443	443	433	0
3	69 (M)	DVT, PE	FLC kappa, t(4;11)	4	Yes	40%	DVT + PE	5	3	3	84	83	83	225
4	58 (M)	HTN, dyslipidemia, smoking	IgA-L, del13(q), del(14)	2	Yes	40%	STEMI	30	1	1	188	214	154	349
5	63 (F)	HTN, dyslipidemia	IgG-K, +1q, -13	4	Yes	0.02%	DVT	10	1	1	171	157	157	287
6	58 (F)	-	FLC-L, IGH rearrangement, t(11q13), +1q	3	No	15%	Hematuria	13	2	1	154	154	159	218
7	55 (F)	HTN, DM2, CHF	IgG-K	4	Yes	20%	Epistaxis	3	0	0	428	428	428	0
8	70 (M)	Smoking	IgA-L, + 1q, -P53 (73%), t(4:14)	4	Yes	8%	Cutaneous bleeding	20	2	2	144	142	142	239

Abbreviations: CAR-Ts: chimeric antigen receptor T-cells; BM: bone marrow; CRS: cytokine release syndrome; F: female; M: male; HTN: hypertension; DM2: diabetes mellitus type 2; NSTEMI: non-ST elevation myocardial infarction; FLC: free light chain (L: Lambda, K: Kappa); ACS: acute coronary syndrome; DVT: deep vein thrombosis; STEMI: ST-elevation myocardial infarction; PE: pulmonary embolism; CHF: congestive heart failure, EMD: extramedullary disease. * Delta fibrinogen refers to the difference between fibrinogen levels at baseline (before CAR-T infusion to the lowest fibrinogen measured post-infusion).

## Data Availability

Detailed data can be provided by the corresponding author upon written request.

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
