# Peer review of "High Rate of Cytokine Release Syndrome-Related Coagulopathy with Low Incidence of Bleeding and Thrombosis in Patients Treated with B-Cell Maturation Antigen (BCMA)-Targeted Chimeric Antigen Receptor T-Cells (CAR-T)"

_cancers, 2025, doi:10.3390/cancers17213551_

Round 1

Reviewer 1 Report

Comments and Suggestions for Authors

This is an academic study characterizing toxicities from BCMA-targeted CAR-T therapy (HBI0101) in 108 adult patients with relapsed/refractory multiple myeloma (RR-MM) and AL amyloidosis. It reports that CRS-related coagulopathy is common. Hypofibrinogenemia (defined as mg/dL) was observed in 73% of patients and was strongly correlated with CRS severity, the number of tocilizumab doses, and increased inflammatory markers (ferritin and LDH). Despite this high rate of laboratory coagulopathy, the study found that clinically relevant complications were infrequent within three months post-infusion, with five thrombotic events (4.6%) and three minor bleeding events (2.7%), and no cases of major or clinically relevant non-major bleeding. The authors conclude that these low rates of complications, especially compared to historical studies, suggest that the prophylactic use of enoxaparin and fresh frozen plasma (FFP) was beneficial.

Although, this study offers some very novel and useful insights, it also suffers from few limitations, namely;

  1. The study is limited by its single-center and observational design
  2. Measurements of key coagulation markers, specifically D-dimer, were not available in the study protocol
  3. Lack of Statistical Power for Clinical Outcomes: Due to the small number of observed thrombotic (n=5) and bleeding (n=3) events, the statistical analysis lacked the power to definitively confirm an association (or lack thereof) between these clinical outcomes and the degree of coagulopathy or CRS severity.

Would be curious to know if the authors are willing to expand this study to multi-center design to provide enough statistical power. Despite this, the manuscript looks good.

Author Response

Thank you again for this positive and constructive review. All the above limitations have indeed been acknowledged in our Results and Discussion sections.

With regards to future research plans: we are currently expanding our study to a larger cohort in patients with multiple myeloma and AL amyloidosis receiving anti-BCMA CAR-T in our center, which includes commercial anti-BCMA products and approaches 200 patients. This study will include measurements of D-dimer, thrombin generation assays (TGA) and scoring of endothelial activation (EASIX), as well as long-term follow-up beyond the three months restriction in the current study. We have an additional cohort of over 200 patients with lymphomas that we are considering adding to the analysis. Collectively these two cohorts are expected to provide robust analyses with statistical power that will improve our understanding of CRS-related coagulopathy, as well as thrombosis and bleeding risks associated with CAR-T, in general. Furthermore, we are in early stages of initiating a multi-center effort with other major CAR-T-treating centres in Israel addressing the above clinical questions. Such multi-centre study will provide a large cohort of patients; however, may be complicated by the non-uniform prophylactic strategies varying between the centres. We have added a sentence to our Discussion section (5th paragraph) acknowledging these efforts.

Reviewer 2 Report

Comments and Suggestions for Authors

Specific comments to the authors

In their submitted manuscript, 'Cytokine Release Syndrome-Related Coagulopathy in B-Cell Maturation Antigen (BCMA)-Targeted Chimeric Antigen Receptor T-Cell (CAR-T) Therapy and the Incidence of Bleeding and Thrombosis', the authors studies the frequency, laboratory findings and clinical correlations associated with cytokine release syndrome (CRS)-related coagulopathy in adult patients with relapsed/refractory(RR) - MM and AL amyloidosis who were treated with BCMA-targeted CAR-T. They performed a prospective analysis of 108 patients with multiple myeloma or light chain amyloidosis who were all treated with the academic anti-BCMA CAR-T HBI0101 in a single-centre trial (NCT04720313).

The authors found that CRS grades 1–3 occurred in 93% of patients, with hypofibrinogenaemia in 73%, including 19% with severe coagulopathy. Fibrinogen levels strongly correlated with CRS severity, tocilizumab dosing, and inflammatory markers. Thrombosis (4.6%) and bleeding (2.7%) were rare and unrelated to CRS or coagulopathy severity. No major bleeding or fatal coagulopathy occurred. Overall, CRS-related coagulopathy was common but rarely led to clinical events, suggesting prophylactic measures may benefit this patient group.

Overall, the manuscript gives some interesting clinical data on CRS-related coagulopathy in adult patients with RR-MM and AL amyloidosis treated with BCMA-targeted CAR-T. The manuscript (including presentation) is mostly comprehensible and convincing. The methods are well described. Although the results and discussion are clear presented, some minor changes could be performed by the authors (see specific comments) to improve the manuscript. In conclusion, the presented data are interesting. After incorporating the mentioned specific comments (see below) the manuscript has the potency to be accepted.

Specific comments:

Title: The title largely describes what the study analysed, but not what it found. Please amend accordingly.

Abstract: Please add the relevant p-values to the abstract.

Material and Methods: Regarding the sentence 'Coagulopathy was defined as a fibrinogen nadir of ≤200 mg/dL'. A threshold fibrinogen level of 100 mg/dL was used to differentiate between severe and mild or moderate coagulopathy.” Please provide a brief explanation of the cut-off values used.

Results:

Table 1: Other markers for CRC and ICANS, such as CRP, ferritin, interleukin 6, LDH and D-dimer, as well as creatinine, transaminases and bilirubin, are missing from the table and subsequent analysis (see Table 2). Furthermore, the authors should add more clinico-pathological details of the MM/AL diagnosis.

Table 2: Please add the P-values for the variables 'blood product support', 'thrombosis events' and 'bleeding events'.

Figure 1: Is there a time-dependent association between the complete blood count (CBC) and coagulation parameters, including prothrombin time (PT), activated partial thromboplastin time (aPTT), fibrinogen levels and CRS grade?

Discussion: Please specify the term 'high rate' in the sentence 'We observed a high rate of CRS-related coagulopathy'. The authors should summarise the findings of other published studies on CRS-related coagulopathy in a separate table for direct comparison of the observations. The authors should update their literature references (see Nat Commun. 2024 Apr 20;15(1):3371. doi: 10.1038/s41467-024-47801-8. PMID: 38643278 or J Clin Oncol. 2023 Sep 20;41(27):4416-4429. doi: 10.1200/JCO.23.00512. Epub 2023 Jul 20.PMID: 37471687 for example as well as others more).

Author Response

Thank you so much for this comprehensive and helpful review, significantly contributing to the quality of the manuscript.

Title: The title largely describes what the study analysed, but not what it found. Please amend accordingly.

Response: The title was amended per the reviewer’s suggestion.

Abstract: Please add the relevant p-values to the abstract.

Response: We’ve added the p-values for the associations between coagulopathy (as reflected via fibrinogen levels) and LDH, ferritin levels and nadir neutrophil count. Following this correction, the abstract is now 258 words long, slightly longer than the recommended length of 250 words, but apparently allowed per the Author Guidelines.

Material and Methods: Regarding the sentence 'Coagulopathy was defined as a fibrinogen nadir of ≤200 mg/dL'. A threshold fibrinogen level of 100 mg/dL was used to differentiate between severe and mild or moderate coagulopathy.” Please provide a brief explanation of the cut-off values used.

Response: The cutoff values used are based on a recent publication by the International Society of Thrombosis and Haemostasis, introducing updated definitions of DIC and detailing several stages of DIC severity. Aside from the definition of overt DIC (which correlates with severe coagulopathy in our study), this publication defines additional “milder”, partly overlapping, entities, namely: non-overt DIC, early-phase DIC, pre-DIC and coagulopathy. Coagulopathy is defined as a “mild to moderate impairment in one or more elements of coagulation”.  This definition is further supported by publications describing hypofibrinogenemia as a key component of coagulopathy, with increased risk of bleeding observed at fibrinogen levels below 200 mg/dL (see references 23 and 24 in the revised manuscript). We have modified the ‘Evaluation of laboratory and clinical parameters’ paragraph in our Methods section to better clarify this matter.

Results:

Table 1: Other markers for CRC and ICANS, such as CRP, ferritin, interleukin 6, LDH and D-dimer, as well as creatinine, transaminases and bilirubin, are missing from the table and subsequent analysis (see Table 2). Furthermore, the authors should add more clinico-pathological details of the MM/AL diagnosis.

Response: We have added the proportion of multiple myeloma patients with extramedullary disease, high-risk cytogenetics and triple-class refractoriness to Table-1. CRP, LDH and ferritin were included in our analysis and appear in Table 2, while IL6 and D-dimer levels were unfortunately not included in the study protocol. Numerical values of creatinine, transaminases and bilirubin were not documented in this coagulopathy-study’s database; however, these were included in the publications reporting the main safety and efficacy outcomes of our academic anti-BCMA CAR-T, referred to in the manuscript (references 7-9).

Table 2: Please add the P-values for the variables 'blood product support', 'thrombosis events' and 'bleeding events'.

Response: The p values for platelet and packed RBC transfusions were added in both Table 2 and the manuscript Results. FFP was administered prophylactically to patients with severe coagulopathy per protocol, hence is a directly dependent variable precluding an association analysis with degree of coagulopathy. With regards to thrombosis and bleeding: p values show no association; however, as explained in the manuscript, the number of events is too small for an adequately powered statistical analysis confirming lack of association. Table 3 was included to provide a case-by-case impression that reflects the apparent lack of association between CRS and/or coagulopathy to thrombosis and bleeding complications.

Figure 1: Is there a time-dependent association between the complete blood count (CBC) and coagulation parameters, including prothrombin time (PT), activated partial thromboplastin time (aPTT), fibrinogen levels and CRS grade?

Response: We initially did not include a time-dependent documentation of peak or nadir CBC and coagulation parameters in our database. We acknowledge that this is a limitation of our study. To respond to this important comment, we returned to the medical records and retrieved additional data. We have added an analysis of time from onset of CRS to initial decrease in fibrinogen and to nadir fibrinogen. See second paragraph under ‘CAR-T associated toxicities’ in Results.

Discussion: Please specify the term 'high rate' in the sentence 'We observed a high rate of CRS-related coagulopathy'.

Response: This was specified.
See first sentence of the Discussion.

The authors should summarise the findings of other published studies on CRS-related coagulopathy in a separate table for direct comparison of the observations.

Response: We have attempted to provide such a table in the original manuscript preparation; however, data with regards to markers of coagulopathy as well as thrombosis and bleeding events in previous studies in multiple myeloma are completely lacking (except for a single study), while in patients with ALL and/or LBCL, reporting was partial and inconsistent. Hence, we felt that the available data is too non-uniform to be organized into a table.

The authors should update their literature references.

Response:  References were updated, with several references added, including the review paper by Holstein et al.

Reviewer 3 Report

Comments and Suggestions for Authors

In this article, the authors contributed to characterizing coagulopathies associated with BCMA-directed CAR-T cell therapy. They described both from a laboratory and clinical point of view the CRS-associated coagulopathy detected in a prospective study on patients with myeloma and amyloidosis, also suggesting prophylactic strategies to be adopted in case of onset of the coagulation disorder. This work is innovative because it improves our understanding of these coagulopathies, the interpretation and management of which still needs to be refined.

The paper is well written and the analysis is very detailed. I would ask the authors to consider the following revisions before acceptance for publication.

  1. The paper does not describe the timing of coagulopathy onset and its relationship with CRS. I suggest reporting the median onset of coagulopathy, its temporal relationship with CRS, and whether there are any clusters from a temporal perspective.
  2. Patients with hypofibrinogenemia following CRS, especially when associated with hyperferritinemia and cytopenias, may be hiding a hemophagocytic syndrome. The authors should specify in the paper how they evaluated this condition, whether they found cases of HS IEC, or whether none of the enrolled patients met the diagnostic criteria.
  3. There are scores directly related to endothelial activation (e.g., EASIX score) that have shown a correlation with the development of coagulopathy and CRS in CAR T studies in LBCL. If such data are available, I suggest that the authors should evaluate whether there is a correlation between endothelial activation and the development of clinical and/or laboratory coagulopathy.

Author Response

  1. The paper does not describe the timing of coagulopathy onset and its relationship with CRS. I suggest reporting the median onset of coagulopathy, its temporal relationship with CRS, and whether there are any clusters from a temporal perspective.

    Response: Thank you for this important comment. Our study was indeed limited by the lack of temporal analysis of CBC and coagulation parameters and their relationship to CRS. In response to this comment, we went back to the medical records and performed a chronological analysis of time from CRS-onset to initial decrease in fibrinogen and nadir fibrinogen levels. See ‘Results’ section in the revised manuscript.

  2. Patients with hypofibrinogenemia following CRS, especially when associated with hyperferritinemia and cytopenias, may be hiding a hemophagocytic syndrome. The authors should specify in the paper how they evaluated this condition, whether they found cases of HS IEC, or whether none of the enrolled patients met the diagnostic criteria.

    Response: Three patients in our cohort indeed fulfilled criteria for IEC-HS. All three patients also exhibited severe coagulopathy occurring concurrently and had grade-2 or higher CRS. This important observation was added to the manuscript in both Methods (‘Evaluation of clinical parameters) and Results (‘CAR-T associated toxicities’ and ‘The association between CRS and coagulopathy’) sections.

  3. There are scores directly related to endothelial activation (e.g., EASIX score) that have shown a correlation with the development of coagulopathy and CRS in CAR T studies in LBCL. If such data are available, I suggest that the authors should evaluate whether there is a correlation between endothelial activation and the development of clinical and/or laboratory coagulopathy.

    Response: Unfortunately, we did not document numerical creatinine levels in our database (we did document occurrence of acute kidney injury), hence our database is insufficient to enable calculation of the EASIX score. This parameter along with others were recently added to the database and study protocol and will be used in future analyses (see also response to Reviewer 1).
